# Emergency Healthcare Providers’ Perceptions of Preparedness and Willingness to Work during Disasters and Public Health Emergencies

**DOI:** 10.3390/healthcare8040442

**Published:** 2020-10-29

**Authors:** Mohammed Ali Salem Sultan, Jarle Løwe Sørensen, Eric Carlström, Luc Mortelmans, Amir Khorram-Manesh

**Affiliations:** 1Directorate of Health Affairs in Najran, Najran 66255, Saudi Arabia; 2Institute of Health and Care Sciences, Sahlgrenska Academy, Gothenburg University, 413 46 Gothenburg, Sweden; eric.carlstrom@gu.se; 3USN School of Business, Campus Vestfold, University of South-Eastern Norway, 3603 Kongsberg, Norway; jarle.sorensen@usn.no; 4Center for Research and Education in Emergency Care, University of Leuven, 3000 Leuven, Belgium; luc.mortelmans@zna.be; 5Regedim, VUB, Brussels and dept of Emergency Medicine, ZNA, Stuivenberg, 2060 Antwerp, Belgium; 6Institute of Clinical Sciences, Sahlgrenska Academy, Gothenburg University, 413 90 Gothenburg, Sweden; amir.khorram-manesh@surgery.gu.se; 7Department of Research and Development, Swedish Armed Forces Centre for Defence Medicine, 426 76 Gothenburg, Sweden

**Keywords:** confidence, disaster, emergency, healthcare, family member, preparedness

## Abstract

This study evaluates the perceptions of preparedness and willingness to work during disasters and public health emergencies among 213 healthcare workers at hospitals in the southern region of Saudi Arabia by using a quantitative survey (Fight or Flight). The results showed that participants’ willingness to work unconditionally during disasters and emergencies varied based on the type of condition: natural disasters (61.97%), seasonal influenza pandemic (52.58%), smallpox pandemic (47.89%), SARS/COVID-19 pandemic (43.56%), special flu pandemic (36.15%), mass shooting (37.56%), chemical incident and bombing threats (31.92%), biological events (28.17%), Ebola outbreaks (27.7%), and nuclear incident (24.88%). A lack of confidence and the absence of safety assurance for healthcare workers and their family members were the most important reasons cited. The co-variation between age and education versus risk and danger by Spearman’s rho confirmed a small negative correlation between education and danger at a 95% level of significance, meaning that educated healthcare workers have less fear to work under dangerous events. Although the causes of unsuccessful management of disasters and emergencies may vary, individuals’ characteristics, such as lack of confidence and emotional distractions because of uncertainty about the safety issues, may also play a significant role. Besides educational initiatives, other measures, which guarantee the safety of healthcare providers and their family members, should be established and implemented.

## 1. Introduction

A disaster is defined as “a serious disruption of functions in a community or a society resulting in widespread human, material, economic or environmental losses which exceed the ability of the affected community or society to cope using its resources” [1,2]. The Centre for Research on the Epidemiology of Disasters (CRED) reported that about 205 million people suffer from the negative outcomes of disasters each year [3]. These numbers of people continue to increase each year, with the increasing in the number of casualties and the severity of disasters such as volcanic eruption, earthquake, storm, flood, etc. [4]. The myriad of disasters and public health emergencies has compelled countries to rethink their security paradigms and preparedness to reduce the number of fatalities and the severity of destruction and disruptions [5].

Disaster preparedness should involve all levels of response systems [6]. The readiness to cope with a disaster encompasses all planning activities which take place at the state, institutional, and individual levels. Countries such as Sweden have maintained a societal security orientation in a bid to safeguard people and property and sustain resilience in planning and preparedness for unforeseen incidents and issues which may cause societal instability [7]. However, a number of stressors and an increased focus on danger may be demotivating factors for all agencies, but in particular healthcare professionals who work to help those in need in times of disasters and emergencies. Therefore, the perceived concerns of healthcare personnel should be highlighted and addressed.

While there are guidelines to assist practitioners in responding to emergency situations, the confidence of healthcare staff might be affected by factors such as individual characteristics and preparedness, training, family support, and communication [7,8]. Several studies have shown that various personal characteristics, such as age, experience, and education, have an effect on or are associated with how individuals perceive the importance of the risks [8,9]. Furthermore, certain factors such as the type of disaster, concern for family, concerns about personal safety, pet care needs and the lack of personal protective equipment (PPE) may influence the willingness (or lack of willingness) of staff to work during disasters [10]. For instance, some staff failed to return to their duties during the Bangkok flooding in 2011, as they were worried about the safety and locations of their relatives [11]. Although hospitals must be functional and appropriately staffed to receive the injured, hospital response plans seem to fail in considering the destruction of the hospital itself and the physical and emotional conditions of the healthcare workers during an incident. Thus, while healthcare workers are expected to be willing to respond to disasters, in reality, they might be reluctant to do so when the situation poses a threat to their safety [12]. 

The success of disaster plans is predicated on the willingness of the responders, whose perceptions of disaster preparedness are influenced by institutional and individual preparedness, experience of previous incidents, and family support [13,14]. A worker who feels physically unequipped and unsafe, or is not mentally prepared to respond to a disaster due to lack of experience may be reluctant to engage in risky attempts to save others, and to handle the challenges associated with emergency responses [10]. Moreover, as several studies have reported, a hospital’s level of preparation also influences the confidence of the employees [13,15,16]. Consequently, while hospitals may be prepared, staff members such as nurses may not be confident in their ability to respond to a disaster.

It is, thus, not clear to what extent healthcare workers feel prepared or perceive the importance of staying prepared in the face of unprecedented disasters. A good understanding of disaster preparedness perceptions among healthcare workers can help considerably in the design and development of operational strategies, including training and exercises on how to respond to different types of disasters in both developed and developing countries [13].

The World Health Organization (WHO) stresses the need for healthcare institutions to prepare their workforce, including nurses and physicians, among others, to ensure preparedness and speed in dealing with different types of emergencies which may occur in the course of their practice [17]. The Kingdom of Saudi Arabia (KSA) has classified the levels of health service according to the Model of Care (MoC) description, which is divided into Activated people, Healthy communities, Virtual care, Primary care, Secondary care, Tertiary care, and Quaternary care. MoC describes a comprehensive care system for meeting health needs, which shifts the focus from curative care to preventive care. Systems of Care will operate less in hospitals and more in people’s homes and communities [18]. Both governmental and private sectors offer healthcare. In the government sector, all levels of health care are provided free of charge [19]. The Ministry of Health (MOH) covers around 80% of the costs, and the remaining costs is covered by other agencies, such as the Armed Forces Military Services (AFMS), Health Services for Royal Commission in Jubal and Yanbua, the Saudi Arabian Oil Company (ARAMCO) Health Services, and others. The private sector provides all levels of health care to Saudi and non-Saudi for a fee [20].

The KSA has recorded a number of disasters such as flooding and fires. Although some measures to increase the degree of preparedness and responsiveness to various types of disasters, such as yearly exercises by many different agencies, have been suggested by the government, a recent fire at Jazan General Hospital revealed response deficiencies and a lack of disaster planning and preparedness [21]. Despite the collective approach in different exercises, a lack of vital skills and competencies may be associated with an inability among the KSA’s healthcare workers to adequately plan and prepare for unpredicted events [22]. A recent study of this group showed that the majority of emergency staff at hospitals in the southern region of the KSA had good theoretical knowledge but insufficient practical knowledge. The study also showed that staffs with greater educational knowledge were more confident to act in emergencies and disaster events [23].

## 2. Aim

The aim of this study was to evaluate healthcare workers’ perceptions of their preparedness and willingness to work during disasters and public health emergencies in the southern region of the KSA.

## 3. Materials and Methods

### 3.1. Study Design

This study employed a quantitative research design. A survey was used to generate a numeric representation of specific scenarios in healthcare. The behavioral characteristics, opinions, and attitudes of various stakeholders in healthcare were analyzed.

### 3.2. Survey

The study employed a validated English-language version of the (Fight or Flight) survey, provided by one of the authors (L.M.). The survey was developed and described in earlier studies [24,25]. The original version of the (Fight or Flight) survey was developed at the Center for Research and Education in Emergency Care (CREEC) of the University of Leuven, Belgium. A multi-scenario survey was developed, as at the time, literature was limited to pandemics. After a pilot study in one Belgian hospital, several disaster medicine experts from the Flemish Disaster Management course (CREEC, emergency nurses, and the military) validated a multi-centric version. This version does not allow studying the association between fear, stress, and emotion, but provides information necessary to establish such an association. It was modified for use in this study and provides a comprehensive analysis of the topic of study since it covers broad areas of disasters and public health emergencies. It works individually to assess several areas of healthcare and delves into in-depth information about the research. The survey is comprised of 60 items distributed between two sections: a demographic section consisting of 12 questions and a scenario section consisting of four scenarios (Willingness to go to work, Knowledge, Risk and Danger). Each scenario contains 12 dimensions, which can influence the working environment, presented as a question and illustrative example.

The willingness to work under a certain condition was marked as yes or no. The condition itself was marked in a predesignated field with 10 different choices: I will work during this incident if:(a)I know my family is safe and taken care of;(b)I am sure good communication lines with my family are available;(c)My boss comes to work as well;(d)I am trained to handle the situation;(e)I get regular updates on the evolution of the incident;(f)Adequate PPE is provided;(g)I get paid extra for it;(h)I can get antivirals (e.g., Tamiflu) for free;(i)I can get antidotes for free;(j)I can get my vaccinations for free (Appendix A).

Furthermore, participants assessed their perceived knowledge in each scenario on a Likert scale by choosing between 1–10 points. One indicates the lowest and 10 the highest grade of perceived knowledge.

### 3.3. Setting

The study was conducted on Thursday 9 July 2020 at 10 MOH hospitals (*n* = 10) in the Najran region, KSA. Najran is located in the southern part of the KSA, where the border is exposed to potential risk of disasters and armed conflict.

### 3.4. Population and Sample

The included participants were healthcare practitioners working in emergency departments (EDs), intensive care units (ICUs), and disaster teams/units, who were willing to participate, of all ages and gender groups. Workers who did not complete the survey, workers who were not present during the study period because of vacation or maternity leave, and personnel at the consultant or managerial level were excluded. All healthcare workers were informed about the study and its goals by the medical affairs administration in each of the hospitals, and informed consent was obtained. Workers were also informed that their participation was voluntary and that they could leave the study whenever they chose to. The participants were randomly chosen from the list of healthcare staff working in each ICU, ED, and disaster team/unit, thus avoiding the bias involved in choosing a specific group. The sample size was set at 250 healthcare practitioners based on the power calculation (Raosoft Inc., Seattle, WA, USA), assuming 4.5% precision with 50% prevalence and a population size of 508 with a 95% confidence interval specified limits.

### 3.5. Data Collection and Processing

The self-completion survey was presented to the participants through the SurveyMonkey website. All data were handled confidentially. Collected data were stored at the research center in each hospital. Healthcare practitioners answered the surveys on a specific research day to prevent response influence. The respondents were asked to provide accurate information. The information provided was subject only to research purposes, and the researcher could not disclose the respondents’ identities at any time, no matter the circumstance.

### 3.6. Ethical Approval

An ethical committee certificate of approval for the study was obtained from the Institutional Review Board at the General Directorate of Health Affairs in the Najran region (IRB Log Number 2020-28 E; date of approval: 7 July 2020).

### 3.7. Statistics

The homogeneity of the items in the subscales of the Fight or Flight survey was analyzed by calculating Cronbach’s alpha using the Statistical Package for the Social Sciences (SPSS) version 20 (IBM, Texas, USA). Cronbach’s alpha was 0.927, showing high internal consistency; according to Brace et al. [26], this value is considered satisfactory. Other results are descriptively presented in actual numbers and percentages.

## 4. Results

### 4.1. Description of the Study Participants

The total number of respondents was 334, but 121 did not complete the survey and were excluded from the study. Data were collected from the remaining 213 participants. The majority of the participants were females and over 50% were nurses. About 53.52% were 25–34 years old (*n* = 114). Some 63% (*n* = 136) had graduated from university. About 70.89% (151) of the participants were married, and 61.5% (*n* = 131) had children. Among the 131 participants who had children, 90 participants indicated that their children lived with them. A total of 120 participants held leadership positions in their organizations, 175 participants (82.94%) had regular contact with patients, and 128 (60.38%) regularly worked in emergency units such as ICUs and EDs. Most of the participants had some kind of training (Table 1).

### 4.2. Willingness to Respond to Disasters and Emergencies

The study participants’ willingness to go to work during disasters and emergencies varied based on the type of condition (Table 2). There was no condition in which all of the participants were willing to work. Notably, the willingness of the participants to work unconditionally varied across conditions. More than 50% of participants were willing to work unconditionally when facing natural disasters such as flooding (61.97%) or when dealing with a seasonal influenza pandemic (52.58%). Over 40% of the respondents were willing to work unconditionally when facing smallpox (47.89%) and SARS/COVID-19 (43.56%). Up to 36.15% were willing to work unconditionally during a special flu pandemic. Ebola outbreaks (27.7%) and biological incidents (28.17%) such as anthrax were the least favorable incidents for working unconditionally. Of the various human-made disasters, a nuclear incident would result in the lowest number of participants willing to work unconditionally (24.88%), followed by a chemical incident and bombing threat (31.92%), a dirty bomb (32.86%), and a mass shooting (37.56%).

Some of the respondents expressed a willingness to work during disasters and emergencies under certain circumstances (Table 3). About 49.77% had demands for facing a special flu pandemic. Sixty-five of the 106 respondents demanded adequate PPE before they would go to work, 18 indicated that they would only go to work if they were properly trained, and eight indicated a willingness to go to work if their families were safe. In the same vein, 45.54% of the respondents indicated that they were willing to go to work in the case of a SARS/COVID-19 pandemic under certain circumstances. Adequate PPE was a requirement of 74 of the 97 who responded, followed by a consideration of the level of training. Additionally, 41.78% would consider going to work after a bombing under certain circumstances. Of these circumstances, the need to ensure that the respondent’s family was safe and taken care of was considered the most pertinent, followed by the need for adequate training to deal with the situation. Seasonal influenza and a dirty bomb were the next two incidents in which respondents were most willing to go to work only under certain circumstances. For seasonal influenza, most respondents would consider going to work only if they were provided with PPE, were adequately trained to deal with the situation, and were assured that their families were safe.

Ensuring that their families were safe and taken care of and that they had an appropriate level of training were the special considerations under which most of the respondents were willing to go to work. Some practitioners were also willing to work under certain circumstances when faced with an Ebola outbreak, smallpox, and a biological incident. For an Ebola outbreak and a biological incident, the most important special considerations were the skills to handle the situation, closely followed by the availability of adequate PPE. When dealing with smallpox, the most important consideration was the availability of adequate PPE, followed by the knowledge to deal with the situation.

Up to 30.52% of the respondents were willing to work under certain circumstances during a chemical incident. Of these circumstances, the knowledge required to deal with the incident was considered the most crucial, followed by the availability of adequate PPE and the knowledge that the respondent’s family was safe. When facing a nuclear incident or a natural disaster, the most important considerations were whether the respondents were adequately trained to deal with the disaster and whether their families were safe.

A biological incident and an Ebola outbreak emerged as the two major disasters for which most respondents expressed serious doubt that they would go to work, 34.74% and 33.33%, respectively. Another 23.47% of the respondents expressed serious doubt about attending work if there were a chemical incident, while only 4.23% expressed serious doubt that they would go to work if there was a natural disaster. Equally, there were few serious doubts about going to work when faced with a seasonal influenza pandemic, SARS/COVID-19, and a special flu pandemic. Some of the participants were sure that they would not go to work in case of disasters and emergencies. About 33.33% of the respondents were certain that they would not go to work if they were required to deal with a nuclear incident, 14.08% if there was a chemical incident, 12.68% if there was either an Ebola outbreak or a mass shooting, 1.88% if there was a seasonal influenza epidemic, and 3.29% if there was a SARS/COVID-19 outbreak.

### 4.3. Participant’s Knowledge of Various Disasters

Table 4 shows a self-rating of the participants’ knowledge of various disasters and emergencies on a scale of 1 to 10. The respondents rated themselves highly on their knowledge of how to deal with SARS/COVID-19, a seasonal influenza pandemic, and smallpox. On the other hand, the respondents gave themselves low ratings on their knowledge of bombing situations, an Ebola outbreak, a dirty bomb, a mass shooting, biological incidents, and terrorist threats.

### 4.4. Risks and Dangers of Various Disasters

The risk and associated danger of the SARS/COVID-19 pandemic occurring during participants’ lifetimes were both at 80%, followed by bombing and terrorist threats at 63% and 69%, respectively. The likelihood of a seasonal influenza pandemic occurring during the study participants’ lifetimes was viewed to be 62%. A similar rating was given by participants’ perceptions of the dangers posed by the disaster to society and the lives and health of individuals. These ratings indicate that seasonal influenza is considered a likely natural disaster with a significant impact on society. The likelihood of mass shootings, such as the Paris shooting, was considered 60%. Its associated danger to society was considered 64% (Table 5).

The risks of a special flu pandemic and smallpox occurring were considered to be 52%. However, a special flu pandemic was viewed as posing more danger than smallpox, namely, 61% compared to 55% for smallpox. It appears that the participants viewed smallpox as unlikely to have a significant impact on society compared to a special flu pandemic. A chemical incident was considered to have a 48% likelihood of occurring. However, its danger to society was considered high, at 59%. In the same vein, the likelihood of a dirty bomb attack during the study participants’ lifetimes was considered to be 47% and its danger 64% if it occurred. Equally, the risk of an Ebola outbreak was considered low at 42%. Nevertheless, the dangers of such an outbreak on society were considered to be 59%. A nuclear accident was considered unlikely to occur, with the respondents rating its risk as 41% and its danger at 64%, making it the third most dangerous disaster in the views of respondents. The respondents viewed a biological incident as the least likely disaster to occur at 40%. Nevertheless, they reported that it would pose a significant danger to society at 55%.

Because age and education were presented as ranks, a non-parametric test, Spearman’s rho was chosen to test the co-variation between age and education versus risk and danger. The test confirmed a small negative correlation between education and danger at a 95% level of significance (Table 6).

## 5. Discussion

In this study, we examined healthcare workers’ perceptions of preparedness and willingness to work during disasters and emergencies in the southern region of the KSA. The primary reason for choosing this area for an evaluation was the continuous exposure of the region to both manmade and natural disasters, and earlier evaluation of their knowledge and competences [23]. The findings in this study indicate that although a high number of participants had training in disaster management, and were supposed to be prepared to respond to one, most of them were unwilling to provide care unconditionally except when dealing with natural disasters and a seasonal influenza pandemic.

Advanced education qualifications have been reported to play a crucial role in the willingness of healthcare workers to participate in disaster and emergency response [27]. In a previous study, examining the staff readiness in managing disasters in this region, the participants were shown to be theoretically well prepared, and particularly those with greater educational knowledge were more confident to act in emergencies and disasters [23]. Most of the participants in this study (63%) also had a university degree and seemed to be well prepared. However, their enthusiasm for participation in various emergencies was limited. In fact, some of them could refuse to work during some events. Thus, being well prepared does not necessarily mean a willingness to act, and the willingness of staff to manage a condition seems to be significantly linked to their disease-related knowledge and experience [8,28,29]. These findings confirm the results of this study, which shows a selective willingness to take part in the management of some of the disasters or emergencies, such as natural disasters and seasonal flu or SARS. The staff seem to have less fear to handle these events and are more familiar with these conditions, maybe due to the KSA´s disaster profile and the earlier epidemics [23].

Gee and Skovdal [30] argued that risk perception plays a role in determining the extent to which frontline health workers were willing to respond in a disaster or an emergency. The fear of personal safety and well-being of colleagues and family are all constraining factors, which distress and influence staff working attitudes during pandemics [31]. Nurses with experience in nursing patients infected with COVID-19 and nurses working in COVID-19 divisions had shown to have low job-retention intentions due to their emotional concerns and fear of becoming infected [32]. Chafee also reported that certain factors, such as the type of disaster, concern for family, pets, and personal safety have an impact on the willingness of staff to work during disasters [10]. These factors combined with special individual characteristics that influence individuals’ risk perception result in an unwillingness to work during specific situations” [8,9]. Thus, frontline healthcare workers could become more confident in dealing with public health emergencies if they have the required knowledge and assurance of their families’ safety [33]. 

While the number of healthcare workers keen to provide care during disasters and emergencies increases under certain circumstances, the findings in this study, presenting staff refusal to go to work in some types of disasters, are of real concern, and arguably, these findings need to be addressed. Anticipating that healthcare workers’ knowledge, age, and other recorded characteristics could be an interesting determinant of their confidence and willingness to work, this study aimed to determine the association between some of these variables. Because age and education were presented as ranks, a non-parametric test, Spearman’s rho, was chosen, and the co-variation between age and education versus risk and danger was tested. The test confirmed a small negative correlation between education and danger at a 95% level of significance, i.e., those with more knowledge have less fear in working under unexpected incidents and are presumably more confident. 

Several studies have shown that disaster preparedness training positively influences the responses of health staff to disasters and emergencies, and identifies the gap in knowledge and disaster preparedness [29,34]. Since knowledge and experience can promote the willingness to participate in the care of victims during hazardous incidents, it is necessary to provide disaster-specific training to healthcare workers to improve their disaster-related knowledge, increase their confidence, and reduce their fears. There should be more focus on multiagency and multi-professional training of all staff, particularly healthcare workers, irrespective of their positions and involvement in patient care, so that they are better equipped to respond collectively to disasters and emergencies.

This study identified challenges facing healthcare workers in the KSA in establishing a functional disaster response system. Lack of education and training might be one significant challenge to a functional disaster response system. However, safety issues (PPE and family safety) are crucial issues, which may threaten the effectiveness of a disaster response system, as most healthcare workers surveyed indicated that they would not respond unless they were sure that their families were safe.

The findings of this study suggest measures, which can be used to increase the competency of healthcare workers in the KSA in order to improve their efficiency and planning knowledge when dealing with emergencies and disasters. Disaster-specific education has been identified as a viable approach for improving the competency of healthcare workers in disaster management. Khorram-Manesh et al. [35] noted that the lack of standardization is a significant barrier to the effectiveness of disaster management courses. Thus, the KSA should consider establishing minimum standards and evaluation metrics to evaluate disaster management skills and training courses. The internet provides a platform for instructional delivery and should be considered to overcome the scarcity of time as a hindrance to the establishment of proper disaster management [36].

## 6. Limitations

This study has a number of limitations, which should be taken into consideration in future research. The survey was extensive; it contained 60 questions, which could be why 121 participants did not complete the survey. The sample consisted overwhelmingly of nurses and physicians working in ICUs, EDs, and disaster teams/units. The small number of included administrators, paramedics/emergency medical technicians, and supportive services workers was not representative of the entire hospital staff. Furthermore, data were collected in the Najran region in southern Saudi Arabia; thus, the results may not be generalizable to all parts of the country. Finally, the number of participants (*n* = 213) distributed in different professions and age groups results in a varying number of participants in each category, and thus, limits the generalization of the results to the population. Future studies should include a larger number of workers from diverse organizations to achieve representative and comprehensive findings.

## 7. Conclusions

The willingness of healthcare workers to respond is selective and depends on the type of disaster or emergency. This is an unexpected consideration for disaster and emergency planners. Among several factors that determine healthcare workers’ willingness to work during disasters and public health emergencies, appropriate knowledge and skills to confidently manage an incident and the assurance of their families’ safety are two decisive factors. Although we could only find a significant correlation between education and willingness to work during emergencies, previous reports have confirmed a significant correlation between education, age, and years of experience and the perception of hazards and fear and consequently willingness to work under threatening circumstances. While unsuccessful management of disasters and emergencies may be the result of organizational shortcomings and resource scarcity, healthcare workers’ lack of knowledge, skills, and confidence and emotional distractions due to uncertainty about their own safety and that of their families may also play a significant role. Besides educational initiatives, which increase staff members’ confidence through knowledge acquisition and skill improvement, other measures, which guarantee their families’ safety and well-being during an emergency, should be established and implemented. Future contingency and disaster plans should include detailed information concerning all these important factors.

## Figures and Tables

**Table 1 healthcare-08-00442-t001:** Demographic data (*n* = 213).

Variable	*n*	%	Variable	*n*	%
Position			Have children		
Supportive Services	17	7.98	Yes	131	61.5
Nurse	116	54.46	No	82	38.5
Administrator	16	7.51	Children living with them		
Physician	47	22.07	Yes	90	42.65
Paramedic/Emergency Medical Technicians (EMT)	17	7.98	No	121	57.35
Age by year			Function		
20–24	7	3.29	Leader	120	56.87
25–34	114	53.52	Executor	91	43.13
35–44	59	27.7	Regular patient contact		
45–54	25	11.74	Yes	175	82.94
55+	8	3.76	No	36	17.06
Gender			Work at emergency units		
Male	97	45.54	Yes	128	60.38
Female	116	54.46	No	84	39.62
Level of education			Training		
Institute	9	4.23	Disaster Management	171	82.61
College	68	31.92	Epidemic/pandemic	85	41.06
University	136	63.85	Chemical incidents	32	15.46
Marital status			Nuclear incidents	9	4.35
Single	62	29.11	Mass casualty incidents	60	28.99
Married	151	70.89			

**Table 2 healthcare-08-00442-t002:** Willingness to go to work during various conditions (*n* = 213).

All Scenarios Affect Your Hospital and Working Area	Yes, Unconditionally		Yes, under Certain Circumstances a–j #		Have Serious Doubts, Probably Not		I Will Certainly Not Respond	
	*n*	%	*n*	%	*n*	%	*n*	%
Natural disaster (e.g., flooding)	132	61.97	63	29.58	9	4.23	9	4.23
Bombing (e.g., terrorist threat)	68	31.92	89	41.78	38	17.84	18	8.45
Seasonal influenza pandemic	112	52.58	83	38.97	14	6.57	4	1.88
Special flu pandemic (e.g., bird flu)	77	36.15	106	49.77	20	9.39	10	4.69
SARS/COVID-19	93	43.66	97	45.54	16	7.51	7	3.29
Ebola outbreak	59	27.70	56	26.29	71	33.33	27	12.68
Smallpox	102	47.89	69	32.39	27	12.68	15	7.04
Chemical incident	68	31.92	65	30.52	50	23.47	30	14.08
Biological incident (e.g., anthrax)	60	28.17	54	25.35	74	34.74	25	11.74
Nuclear incident	53	24.88	48	22.54	41	19.25	71	33.33
Dirty bomb	70	32.86	83	38.97	37	17.37	23	10.80
Mass shooting (e.g., Paris)	80	37.56	67	31.46	39	18.31	27	12.68

**Explanation of ‘Under certain circumstances’ a–j #:** (a) If I know my family is safe and taken care of; (b) If I am sure good communication lines with my family are available; (c) If my boss comes to work as well; (d) If I am trained to handle the situation; (e) If I get regular updates on the evolution of the incident; (f) If adequate PPE is provided; (g) If I get paid extra for it; (h) If I can get antivirals (e.g., Tamiflu) for free; (i) If I can get antidotes for free; (j) If I can get my vaccinations for free.

**Table 3 healthcare-08-00442-t003:** Willingness to go to work under certain circumstances (see explanations of a–j above).

All Scenarios Affect Your Hospital and Working Area	a	b	c	d	e	f	g	h	i	j	Mean	Std. Deviation
Natural disaster *n* = 63	26	5	2	22	3	2	1	0	1	1	2.97	2.130
Bombing *n* = 89	47	7	3	20	3	7	2	0	0	0	2.56	1.875
Seasonal influenza pandemic *n* = 83	9	2	1	13	1	50	2	0	2	3	5.25	2.118
Special flu pandemic *n* = 106	8	0	2	18	4	65	1	2	2	4	5.48	1.942
SARS/COVID-19 *n* = 97	4	0	2	11	3	74	1	1	0	1	5.56	1.354
Ebola outbreak *n* = 56	4	3	1	23	1	19	2	1	1	1	4.75	1.919
Smallpox *n* = 69	4	2	2	19	1	35	1	0	1	4	5.29	2.108
Chemical incident *n* = 65	13	0	1	25	2	21	2	0	1	0	4.23	1.951
Biological incident *n* = 54	3	1	2	24	2	17	2	1	0	2	4.87	1.914
Nuclear incident *n* = 48	14	0	0	19	4	9	1	0	1	0	3.67	2.077
Dirty bomb *n* = 83	39	2	3	29	2	8	0	0	0	0	2.72	1.783
Mass shooting *n* = 67	27	6	0	23	5	5	0	0	0	1	2.94	2.007

**Table 4 healthcare-08-00442-t004:** Participants’ self-ratings of their knowledge on a Likert scale of 1 (no knowledge at all) to 10 (knowledge on specialist level).

All Scenarios Affect Your Hospital and Working Area	Mean	Std. Deviation	Confidence Interval Lower Bound–Upper Bound
Natural disaster	5.59	2.281	5.28–5.90
Bombing	4.89	2.598	4.54–5.24
Seasonal influenza pandemic	6.77	2.298	6.46–7.08
Special flu pandemic	5.71	2.422	5.39–6.04
SARS/COVID-19	7.80	2.172	7.51–8.10
Ebola outbreak	4.64	2.719	4.27–5.01
Smallpox	6.10	2.490	5.76–6.43
Chemical incident	4.97	2.573	4.62–5.32
Biological incident	4.38	2.711	4.02–4.75
Nuclear incident	3.99	2.705	3.62–4.35
Dirty bomb	4.97	2.507	4.63–5.31
Mass shooting	5.23	2.413	4.91–5.56

**Table 5 healthcare-08-00442-t005:** Description of risk (0–100% danger when it happens during respondent’s lifetime) and danger (0–100% danger disturbs the whole society and threatens population’s lives and health).

All Scenarios Affect Your Hospital and Working Area	Risk	Danger
	%	%
Natural disaster	58	64
Bombing	63	69
Seasonal influenza pandemic	62	62
Special flu pandemic	52	61
SARS/COVID-19	80	80
Ebola outbreak	42	59
Smallpox	52	55
Chemical incident	48	59
Biological incident	40	55
Nuclear incident	41	64
Dirty bomb	47	64
Mass shooting	60	64

**Table 6 healthcare-08-00442-t006:** Ranks and statistics of Age and Education/Risk and Danger (*n* = 213).

Variable	Variable	Age	Risk	Variable	Variable	Education	Risk
Age	Correlation Coefficient	1.0	−0.105	Education	Correlation Coefficient	1.0	−0.044
Sig. (2-tailed)		0.13	Sig. (2-tailed)		0.52
Risk	Correlation Coefficient	−0.105	1.0	Risk	Correlation Coefficient	−0.044	1.0
Sig. (2-tailed)	0.13		Sig. (2-tailed)	0.52	
**Variable**	**Variable**	**Age**	**Danger**	**Variable**	**Variable**	**Education**	**Danger**
Age	Correlation Coefficient	1.0	−0.106	Education	Correlation Coefficient	1.0	−0.162
Sig. (2-tailed)		0.12	Sig. (2-tailed)		0.02
Danger	Correlation Coefficient	−0.106	1.0	Danger	Correlation Coefficient	−0.162	1.0
Sig. (2-tailed)	0.12		Sig. (2-tailed)	0.02

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
