# Peer review of "Emergency Healthcare Providers’ Perceptions of Preparedness and Willingness to Work during Disasters and Public Health Emergencies"

_healthcare, 2020, doi:10.3390/healthcare8040442_

Round 1
Reviewer 1 Report
Dear Authors,
Thank you for an interesting paper timely written for the current pandemic we are currently facing worldwide. I have few comments as below:
- Abstract is very brief. Interesting results (figures/ percentage) should be specifically highlighted in the abstract instead of mentioning "a large nu,mber", "most" etc.
- Line 49-51 is a very short paragraph. May consider combine with the paragraph above or below.
- Method is very briefly described. Needs to elaborate the lowest and highest scores for each domain being analysed
- Would the results be different is the sample size consist of equally distributed respondent from each categories?
- Does age group play an importat role when the respondents give their response towards the % of risk and danger they are currently facing (respondents for each age group is not equally distributed).
- Results:
- Demographic data should be presented in a table format for better readabilty annd understanding for the readers. Highlight important findings only in the text.
- It seems that table 1 and figure 1 reported the same results and these findings also have been described in the text (line 167- 210).
- Most results are descriptive analysis only.
- Discussion
- Lack of critical analysis of the results
Author Response
Dear reviewer,
We would like to thank you for your comments. We have replied to reviewer’s questions and hope that our answers are satisfactory for publishing the paper.
Best wishes
Authors

Reviewer 2 Report
This is a well-written and very interesting article to read. The aim is clear and the reasons for the research are well justified. However, several issues need to be clarified/addressed:
-In the introduction, it is not clear if KSA health professionals must have training on emergency or disaster situations. I recommend the authors to provide some information on the KSA health system in general and in this issue in particular as this may have implications on the conclusions.
-The fight or flight response (in which the survey was used by the authors) should be better explained in why it plays a critical role in how we deal with stress and danger in our environment. This is useful to support not only the methodological part of the paper but also the conclusions (ie the importance of learning more about the professionals’ reaction to stress situations).
-As it is, the 4.1 section is rather confusing. I recommend the authors to present the information in a table.
-The analysis performed is just descriptive. Though interesting, it is rather limited. Correlations between variables should be added. In particular, the characteristics of the participants were not used to analyze differences that may be very useful in a public policy or management perspective. This is a major handicap of the research analysis held by the authors.
-The conclusion’s section is rather limited. The authors should ponder more on what to the with the research results. For instance, the authors should discuss a little bit more if, according to the results achieved, it is expected to have sufficient prepared professionals in crises in KSA and, if not, what needs to be done.
- Finally, figures 1 and 2 do not provide additional information to the tables nor are mentioned in the text.
Author Response

(The authors gave the same response as above.)

Round 2
Reviewer 2 Report
The authors have satisfactorily responded to most of my questions and made the necessary changes to the manuscript.